# CO$_2$ physiological effect can cause rainfall decrease as strong as large-scale deforestation in the Amazon

Gilvan Sampaio[1], Marília Shimizu[1], Carlos A. Guimarães-Júnior[1], Felipe Alexandre[1], Marcelo Guatura[1], Manoel Cardoso[2], Tomas F. Domingues[3], Anja Rammig[4], Celso von Randow[2], Luiz F. C. Rezende[2], David M. Lapola[5]

[1] Centro de Previsão de Tempo e Estudos Climáticos, Instituto Nacional de Pesquisas Espaciais, Cachoeira Paulista SP, 12630-000, Brazil

[2] Centro de Ciência do Sistema Terrestre, Instituto Nacional de Pesquisas Espaciais, São José dos Campos SP, 12227-010, Brazil

[3] Departamento de Biologia, Universidade de São Paulo, Ribeirão Preto SP, 14040-901 Brazil

[4] Land Surface-Atmosphere Interactions, Technical University of Munich, Freising, 85354, Germany

[5] Centro de Pesquisas Meteorológicas e Climáticas Aplicadas à Agricultura, Universidade Estadual de Campinas, Campinas SP, 13083-886, Brazil

*Correspondence to*: David M. Lapola (dmlapola@unicamp.br)

**Abstract.** The climate in the Amazon region is particularly sensitive to surface processes and properties such as heat fluxes and vegetation coverage. Rainfall is a key expression of the land surface-atmosphere interactions in the region due to its strong dependence on forest transpiration. While a large number of past studies have shown the impacts of large-scale deforestation on annual rainfall, studies on the isolated effects of elevated atmospheric CO$_2$ concentrations (eCO$_2$) on canopy transpiration and rainfall are scarcer. Here, for the first time, we systematically compare the plant physiological effects of eCO$_2$ and deforestation on Amazon rainfall. We use the CPTEC-Brazilian Atmospheric Model (BAM) with dynamic vegetation under a 1.5xCO$_2$ experiment and a 100% substitution of the forest by pasture grasslands, with all other conditions held similar between the two scenarios. We find that both scenarios result in equivalent average annual rainfall reductions (Physiology: -257 mm, -12%; Deforestation: -183 mm, -9%) that are above the observed Amazon rainfall interannual variability of 5%. The rainfall decreases predicted in the two scenarios are linked to a reduction of approximately 20% in canopy transpiration but for different reasons: the eCO$_2$-driven reduction of stomatal conductance drives the change in the Physiology experiment, and the smaller leaf area index of pasturelands (-72% compared to tropical forest) causes the result in the Deforestation experiment. The Walker circulation is modified in the two scenarios [Physiology: a humidity-enriched free troposphere with decreased deep convection due to the heightening of a drier and warmer (+2.1°C) boundary layer; Deforestation: enhanced convection over the Andes and a subsidence branch over east Amazon without considerable changes in temperature (-0.2°C in 2-m air temperature and +0.4°C in surface temperature)], but again, these changes occur

through different mechanisms: strengthened west winds from the Pacific and reduced easterlies entering the basin in the Physiology experiment, and strongly increased easterlies influence the result of the Deforestation experiment. Although our results for the Deforestation scenario agree with the results of previous observational and modelling studies, the lack of direct field-based ecosystem-level experimental evidence regarding the effect of $eCO_2$ on moisture fluxes in tropical forests confers a considerable level of uncertainty to any projections of the physiological effect of $eCO_2$ on Amazon rainfall. Furthermore, our results highlight the responsibilities of both Amazonian and non-Amazonian countries to mitigate potential future climatic change and its impacts in the region, driven either by local deforestation or global $CO_2$ emissions.

## 1 Introduction

Despite the agreed upon increase in temperature projected for the tropics in the next decades, future precipitation patterns for the region are still highly uncertain, even regarding anomaly signals (IPCC, 2013). Such uncertainties are particularly relevant for the Amazon region, given not only its dependence on small-scale convection but also the strong dependence of the region's climate on surface processes (Kooperman et al., 2018). It is long known that moisture recycling is a key process in the functioning of the Amazonian system (Eltahir and Bras, 1994), with recycled precipitation reaching values up to 80% in the western part of the basin (Spracklen et al., 2012; Zemp et al., 2017). As such, alterations in the land surface cover, properties and dynamics are expected to drive changes in regional climatic patterns.

Past modelling exercises have shown that large-scale clear-cut deforestation of the Amazon and the substitution of forested lands with pasture or soybean cultivation are associated with substantial changes in the surface Bowen ratio and in the surface temperature, from -0.5°C to +3.1°C, with an accompanying reduction in the provision of humidity to the atmosphere through evapotranspiration and changes in regional atmospheric circulation and convection, with a rainfall reduction of approximately 25% (in the projections where 100% of the forest is substituted by pastures) (Feddema et al., 2005; Lawrence and Vandecar, 2015; Lejeune et al., 2015; Nobre et al., 1991; Sampaio et al., 2007; Spracklen and Garcia-Carreras, 2015). The study conducted by Lorenz et al., (2016) shows the importance of the considered scale of deforestation and whether adjacent areas–which experience increases in horizontal moisture advection–are considered or not. Other studies have covered the multidirectional dynamic feedbacks between the climate and the resilience of the forest, showing the importance of determining the role of the background climate in which deforestation occurs (Li et al., 2016) and the oceanic circulation patterns (Cox et al., 2004; Nobre et al., 2009) when assessing any changes in the vegetation-climate equilibrium in the Amazon region. There is now modelling evidence even regarding the teleconnections of such a climate change caused by Amazon deforestation, that results, for example, in reduced precipitation in the northwest U.S. through the propagation of Rossby Waves (Lawrence and Vandecar, 2015; Medvigy et al., 2013).

Recent studies are now focusing on how more subtle changes in forest dynamics can potentially affect the climate in the Amazon region and elsewhere. Splitting up the effects of increased atmospheric $CO_2$ ($eCO_2$) into its physiological effects on vegetation (the so-called ß sensitivity factor) and the sensitivity of the climate to $eCO_2$ (α) and, thereafter, the impact of the

climate on vegetation (γ), unveils the extent to which the future climate in the Amazon will be controlled by ecophysiological processes or by physical processes (Betts et al., 2007; Cao et al., 2010; Kooperman et al., 2018). The work by Kooperman et al. (2018), for example, shows that the ß effect alone drives a stronger reduction in precipitation in the Amazon region (12%) than the γ effect alone (5%). Such precipitation reduction associated with the ß effect is driven primarily by the reduced stomatal conductance resulting from $eCO_2$ in the employed Earth system model (CESM; (Lindsay et al., 2014)). Therefore, despite the persistence of Amazon forest vegetation in these simulations, the flux of moisture from the land surface to the atmosphere is considerably altered, as in the large-scale deforestation modelling exercises. Notwithstanding, there is no set of coupled land surface-atmosphere simulations that have assessed both the isolated ß and large-scale deforestation effects on climate using the same model(s) with identical boundary conditions.

Here, we perform and systematically compare coupled model simulations on the feedbacks between Amazon forest vegetation and the regional climate, driven either by the physiological effects of $eCO_2$ on vegetation or by large-scale Amazon deforestation with substitution by pastures. Such an exercise allows the timely comparison of the ecophysiological and physical mechanisms involved in the resulting climatic changes predicted in both land surface change scenarios, considering that these mechanisms have thus far been assessed separately in the literature (e.g., (Langenbrunner et al., 2019)). Moreover, the present study also provides baseline hypotheses to be tested in the upcoming free-air $CO_2$ enrichment (FACE) experiment in the central Amazon (Norby et al., 2016). Furthermore, it ultimately draws a timely comparison between the climatic impacts of local direct anthropogenic disturbances such as deforestation, which is of well-determined responsibility and is thus more feasible to resolve (Nepstad et al., 2014), and a global indirect "disturbance" such as $eCO_2$, which has diffuse responsibility and is proving much harder to abate.

## 2 Methods

### 2.1 Climate models

This study is focused mostly on the application and analysis of results obtained from the CPTEC-BAM coupled dynamic vegetation-atmosphere model. The CESM model is employed as a parallel model to specifically test the effects of deforestation and compare its results to those of other studies that employed this model to evaluate the physiological effects of $eCO_2$ on Amazon rainfall (e.g., Kooperman et al. 2018).

CPTEC-BAM is a global atmospheric model created by the Centre for Weather Forecast and Climatic Studies (CPTEC) from Brazil's National Institute for Space Research (INPE), with a horizontal spectral grid T62 (~ 1.875° lat × 1.875° lon) and 28 vertical levels (hybrid sigma-pressure coordinates, with sigma close to the surface and pressure at the top of the atmosphere). Previous studies (e.g., (Cavalcanti et al., 2002; Marengo et al., 2003)) showed that this model was able to simulate the main climatic features of South America, although some systematic errors still remain, such as wet biases over the Andes. The land surface component of CPTEC-BAM is the Integrated Biosphere Simulator (IBIS) (Foley et al., 1996; Kucharik et al., 2000). The model simulates the coexistence of both grass and tree plant functional types (PFTs) in grid cells,

and disturbances such as fires are represented by a fixed percentage of the biomass of all PFTs that is reduced each year. The estimation of stomatal conductance ($g_s$) in IBIS is based on the model by Ball & Berry (1982) with an equation (Eq. 1) that describes the response of $g_s$ to the carbon assimilation rate ($A_n$), relative humidity ($h_s$) and atmospheric $CO_2$ concentration ($c_s$) (Collatz et al., 1991):

$$g_s = m\frac{A_n h_s}{c_s} + b, \tag{1}$$

where $m$ and $b$ are the slope and intercept coefficients, respectively, and are obtained by analysing the linear regression of leaf gas exchange data in an environment with controlled ventilation and temperature (Ball et al., 1987). The coefficient $m$ has values of 11 and 4 for tropical evergreen forest and tropical ($C^4$) grasslands, respectively. The coefficient $b$ has a value of 0.01 for tropical evergreen forest and a value of 0.04 for $C^4$ grass. Hydraulic stress control over stomatal conductance is

105 considered through the incorporation of a multiplying parameter based on soil water moisture, ranging from 0 to 1.

CESM is an Earth system model developed by USA's National Center for Atmospheric Research (NCAR) that provides simulations of the Earth's climate (Hurrell et al., 2013). CESM is composed of five separate models representing the Earth's atmosphere (Community Atmosphere Model version 5-CAM5), ocean (Parallel Ocean Program-POP version 2), land (Community Land Model 4.5-CLM4.5), land-ice (Glimmer ice sheet model-G- CISM), and sea-ice (Community Ice CodE-

110 CICE4) systems. These components communicate with each other through a central coupler component. The CESM system allows several resolution configurations and combinations of components and includes the potential for making simulations with only the surface component or with the surface component coupled with the atmospheric model, among many other combinations. The spatial resolution used is 0.9° lat x 1.25° lon or approximately 100 km.

**2.2 Modelling protocol**

The numerical experiments employed here include simulations that consider the increase in the concentration of atmospheric $CO_2$ affecting plant physiology as well as experiments that consider deforestation in the Amazon, as follows (Table 1):

Control: Control runs with an atmospheric $CO_2$ concentration of 388 ppmv, one with a dynamic and another with a static geographical distribution of vegetation types (for comparison with the Physiology and Deforestation scenarios, respectively).

Physiology: Sensitivity run with a $CO_2$ concentration of +200 ppmv, equivalent to an increase of 1.5x from the control $CO_2$

value. This concentration affects only plant physiology and not the radiative balance of the atmosphere.

Deforestation: Sensitivity run with deforestation of the Amazon, wherein the original forest cover is 100% replaced by $C^4$ grass pasturelands (Fig. 1).

RCP8.5+Def: Sensitivity run using RCP8.5's $CO_2$ increase trajectory affecting both plant physiology and the atmospheric radiative balance, with concomitant replacement of 100% of forest cover by $C^4$ grass pasturelands (the results of which are

125 shown in this article's supplement).

The selection of such scenarios starts with the intention of understanding the impacts on moisture fluxes and rainfall in the Amazon that are driven by the target concentration to be used in the AmazonFACE experiment in the central Amazon

(Norby et al., 2016). Second, we also wanted to know how the results obtained in the Physiology scenario compared to the changes expected due to extreme deforestation in the region. Rather than representing realistic projections of the future of the Amazon, this systematic separation of climatic forcing types allows us to better understand how each forcing contributes to future changes in the region. Notwithstanding, an atmospheric $CO_2$ concentration of +200 ppm (i.e., 588 ppm) is projected to be reached shortly after 2050 under the IPCC RCP8.5 scenario and in 2080 under the RCP6.0 scenario (Vuuren et al., 2011). Complete deforestation of the Amazon basin, following a business-as-usual deforestation-rate scenario–with deforestation rates typical of the late 1990s–could possibly be reached in approximately 2100 (Soares-Filho et al., 2006).

For all model runs, sea surface temperature was considered the climatological mean annual cycle for the 1981-2010 period. In the experiments with increasing $CO_2$, the dynamic vegetation scheme was turned on, meaning that the geographical distribution of vegetation types could vary throughout each model run according to the variations in the climatic variables (given that our analysis is focused on precipitation patterns over the Amazon region, the dynamic vegetation changes are not analysed here, especially because there are no significant changes from broadleaf forest to other vegetation types in the $eCO_2$ runs). On the other hand, dynamic vegetation is disabled and $C^4$ grass vegetation was prescribed and held constant until the end of the integration in the experiments representing the deforestation of the Amazon rainforest. The numerical experiments with dynamic vegetation were integrated for a period of 100 years, with constant $CO_2$ concentrations as prescribed in Table 1. Both the control and sensitivity runs for the Deforestation scenario were run for a period of 30 years, given that these runs employed static vegetation. All control and experimental simulations were carried out using three different initial conditions derived from three distinct days (Jan 1, 2003; Oct 10, 2007; Dec 17, 2017) of US' National Centers for Environmental Prediction (NCEP) reanalysis. The analysis of all scenarios relied on averaged results over the last 30 10 years of each simulation.

Similar "Control" and "Deforestation" experiments were carried out using the CESM model for a comparison with the "Physiology" runs conducted using this model in other studies (Cao et al., 2010; Kooperman et al., 2018). These CESM simulations were configured with only the atmospheric and land surface components enabled to produce simulations that could be comparable with those of CPTEC-BAM.

## 3. Results

Fig. 2 shows that both $eCO_2$ and deforestation are associated with considerable reductions in precipitation across the Amazon region, especially in the eastern and central Amazon regions in the Physiology and Deforestation scenarios conducted with CPTEC-BAM. Two remarkable differences between the Physiology and Deforestation runs regarding the spatial patterns of precipitation changes are the extension of the reduction area over Bolivia and south Peru in the latter model run and the strong localized precipitation increase over Colombia and Venezuela in the former model scenario. In fact, the average precipitation reduction estimated with CPTEC-BAM is stronger in the Physiology run than in the Deforestation scenario, with decreases of -0.70 mm d$^{-1}$ and -0.50 mm d$^{-1}$, respectively, which represent 12% and 9% of the

region mean annual precipitation; however, the ranges of variation of the anomalies in both scenarios do not indicate a significant difference between the two mean values (Fig. 3a).

As expected for a tropical region where variations in precipitation and temperature are tightly coupled, reductions in evaporative cooling and changes in atmospheric circulation are combined with changes in the regional near-surface air temperature: +2.1°C in the Physiology scenario and -0.2°C in the Deforestation scenario (Fig. 3b). Although the predicted changes in the moisture budget are similar between these two scenarios (Table 2), we attribute the moderate change in the near-surface atmospheric temperature and the decrease in the sensible heat observed in the Deforestation scenario as the results of a strong increase in near-surface atmospheric advection (see section 3.2). Part of the observed evapotranspiration decrease in the Deforestation scenario is also a result of the increase in albedo (from 0.13 to 0.19). Notwithstanding, the substitution of forest by pastures reduces the transference of humidity from the surface to the atmosphere, driving a decrease in latent heat that is comparable to that also observed in the Physiology run. The reductions in evapotranspiration (Physiology: -0.35 mm d$^{-1}$; Deforestation: -0.28 mm d$^{-1}$) are associated with reductions in moisture convergence [precipitation minus evapotranspiration (Banacos and Schultz, 2005)] alongside decreased precipitation in both the Physiology and Deforestation model scenarios. The reduction in moisture convergence is 59% more pronounced in the Physiology scenario (Fig. 3a) than in the Deforestation scenario due to the strong reduction in the horizontal transport of humidity by easterly winds. The mechanisms associated with these changes are explained in the next sections.

### 3.1 Provision of humidity

The similarity of the changes in average precipitation, evapotranspiration and moisture convergence between the Physiology and Deforestation scenarios reveals the strength of the forest's ecophysiological (*i.e.,* stomatal) control on the regional climate (Fig. 4). The effect that a higher $CO_2$ concentration has on reducing $g_s$ (Eq. 1) overcomes the positive effect of increased gross primary productivity (GPP) (Physiology: +7.0 $\mu molCO_2$ m$^{-2}$ s$^{-1}$ (+58%); Deforestation: -1.0 $\mu molCO_2$ m$^{-2}$ s$^{-1}$ (-16%) on $g_s$, resulting in a net reduction in stomatal conductance in the Physiology run of -0.10 mol m$^{-2}$ s$^{-1}$ (-26%), related to a decrease of -0.35 mm d$^{-1}$ (-18%) in canopy transpiration (Table 2). On the other hand, the decreases in precipitation and evapotranspiration obtained in the Deforestation run (Fig. 4) do not result in the considerably lower $g_s$ that is generally maintained by C$^4$ grasses (-0.02 mol m$^{-2}$ s$^{-1}$; -4%). However, the considerable reduction in the leaf area index (-72%) and a slightly decreased GPP are associated with an average decrease in transpiration (-0.42 mm d$^{-1}$; -22%) in the Deforestation scenario). Notwithstanding, a counterintuitive increase in specific moisture along the vertical atmospheric profile above the planetary boundary layer is found in the Physiology model run with CPTEC-BAM (+0.32 g kg$^{-1}$), whereas the same model shows a decrease in specific humidity in the Deforestation run (-0.07 g kg$^{-1}$) (Fig. 5b and d).

### 3.2 Atmospheric circulation

As previously modelled in the study by Kooperman et al. (2018) using CESM, eCO$_2$ is related to convective heating over Central Africa that drives anomalous eastward flows across the tropical Atlantic Ocean, ultimately affecting the flow of

humidity into the Amazon basin (Fig. 5). In fact, there is also a strengthening of the Walker cell observed in CPTEC-BAM over the Amazon region, with increased moisture convergence in north South America (also helped by stronger westerlies from the Pacific in this region) that is not as strong as that observed in in CESM but is sufficient to result in a precipitation

increase in the north Andes and an atmospheric stabilization with precipitation decreases across most of the Amazon.

The atmospheric circulation changes are completely different in the Deforestation scenario (Fig. 5c), in which there is a pronounced increase in easterlies across the entire Amazon region as a result of decreased roughness length of surface vegetation [2.65 m in tropical evergreen forest and 0.08 m in $C^4$ grass (Sampaio et al., 2007)] and the reduced pumping of deep soil moisture to the atmosphere, especially in the dry season (June to October) (Fig. 6d). Fig. S5 shows the meridional

mean planetary boundary layer height at the equator over the Amazon under the different scenarios. In the Deforestation scenario, there is an average decrease of 10% in the boundary layer height, attributable to the considerably lower surface roughness length of pastures compared to that of tropical forests. On the other hand, there is an average increase of 21% in the boundary layer height in the Physiology run, associated with the increased heating of the surface. As a result, $eCO_2$ causes a higher, drier and warmer boundary layer over the Amazon that acts as a barrier to a humidity-enriched, though

shallower, simulated free troposphere with less deep convection (c.f. Langenbrunner et al., 2019). On the other hand, the strong increase in westward moisture advection, aligned with the increased albedo and decreased vertical mixing (Fig. S5) seems to best explain the nearly unchanged surface temperature seen in the Deforestation scenario. The superposition of the spatial pattern of changes in moisture convergence over the 850-hPa atmospheric circulation anomalies shows that different circulation patterns produce similar changes in the region's atmospheric moisture budget (Fig. 5a and 5c).

The reduction in latent heat flux in our simulations (Fig. 3c and Table 2) also helps reduce convection over the Amazon region, tending to cool the upper atmosphere and reinforce atmospheric stabilization.

These changes in horizontal circulation imply, in the Physiology scenario, that less moisture enters the Amazon region from the Atlantic (-4.9 kg $m^{-1}$ $s^{-1}$) and less moisture leaves the regions towards the Andes (-2.6 kg $m^{-1}$ $s^{-1}$) (this latter is somewhat compensated by a stronger moisture convergence from the Pacific to the Andes, as shown in Fig. 5b). In the Deforestation

scenario, there is an increase in the input of humidity to the Andes at the surface level (on the order of 3.0 kg $m^{-1}$ $s^{-1}$), which is also perceptible in the western part of the vertical humidity profile near the surface levels (Fig. 5d). The lower evapotranspiration capacity aligned with the lower vertical mixing due to pasture's lower roughness length (than that of forests) results in an atmospheric volume that is depleted of moisture and shows a decreased uplifting of air masses. In the Physiology scenario, despite the decreased evapotranspiration capacity, the increased surface heating increases vertical

mixing at low levels (up to 700 hPa), associated with a deeper boundary layer and higher mixing layer, which is, in turn, connected to the increase in humidity throughout the free tropospheric volume (above the boundary layer) over the region. However, after such atmospheric heights, there are strong subsidence anomalies seen in the Physiology run (Fig. 5b), which decrease deep convection that is ultimately associated with lower rainfall rates. The same vertical circulation patterns have been demonstrated well in previous (separate) studies that modelled the large-scale deforestation of the Amazon and, more

recently, the isolated physiological effects of $eCO_2$ on the region's climate (*c.f.* Langenbrunner et al. 2019).

### 3.3 Radiative balance

A decrease in the surface sensible heat ($-1.34$ W m$^{-2}$) in the Deforestation run (Fig. 3c), alongside a decrease in the latent heat, results in a negative net surface radiation balance in the Deforestation run, associated with a small decrease in the average 2-m air temperature ($-0.2°C$) (Table 2) (but also with an increase of $+0.4°C$ in surface temperature). On the other hand, in the Physiology scenario, an increase in sensible heat ($+3.96$ W m$^{-2}$) is observed, associated with an average increase in the 2-m air temperature of $+2.1°C$. While the decrease in latent heat is also directly connected to a lower evapotranspiration capacity, the opposite results shown in each scenario regarding sensible heat are also associated with opposite changes in near-surface atmospheric circulation patterns: in the Deforestation run, there is an increase in near-surface atmospheric advection, whereas in the Physiology scenario, this advection is considerably decreased (as explained in section 3.2 Atmospheric circulation). Shortwave radiation is increased due to decreased nebulosity in both model scenarios (Physiology: $-1.4\%$; Deforestation: $-2.2\%$), but such an increase in the shortwave radiation balance is stronger in the Deforestation scenario due to the albedo change. The same pattern is also obtained for the surface balance of longwave radiation, which increases in both scenarios but increases more strongly in the Deforestation run (Physiology: 2.7 W m$^{-2}$; Deforestation: 6.9 W m$^{-2}$), which is probably a combination of the lower evapotranspiration capacity and increased horizontal advection in the latter scenario.

### 3.4 Seasonality

Precipitation is consistently below the control values year-round in the Physiology and Deforestation experimental model runs (Fig. 6a and S4a). However, differences regarding monthly precipitation between the Physiology and Deforestation scenarios are evident at the end of the dry season and at the onset of the rainy season (August to December). In this regard, precipitation seasonality is stronger in the Deforestation scenario than in the Physiology model run. This is closely linked to changes in evapotranspiration given that the permanence of the forest in the Physiology scenario supports a higher evapotranspiration flux during the dry season compared to that in the Deforestation run (Fig. 6c). On the other hand, the evaporation values in the Deforestation run are, for most of the year, above the control values, which explains the higher evapotranspiration observed during the rainy season in comparison to that seen in the Physiology scenario (although evapotranspiration is reduced in comparison to the control run, following the reduction in precipitation).

As shown, for example, by Kooperman et al. (2018), the physiological effects of eCO$_2$ on the region's climate take place, namely, in the wet season, when GPP is higher and transpiration is lower (see Fig. 6d and h), even though our results also show a considerable rainfall reduction during the dry season. Conversely, it has been demonstrated (e.g., by Lawrence & Vandecar 2015) that large-scale deforestation causes climatic changes specifically during the dry season, when transpiration is particularly reduced, as was also shown in our results (Fig. 6a and d).

These seasonal variations in evapotranspiration are at least partly explained by the opposing seasonal patterns of canopy transpiration in the Physiology and Deforestation scenarios (Fig. 6d). On the one hand, the highest values of this variable in the Physiology run occur during the dry season, when a high vapour pressure deficit increases the evapotranspiration demand that trees can fulfil (at least partially) even under the given $eCO_2$. On the other hand, the lowest canopy transpiration values in the Deforestation run occur during the dry season as a result of seasonal decreases in the pasture leaf area index and root depth in this scenario.

Stomatal closure driven by $eCO_2$ is related to higher water use efficiency [the amount of water used (in transpiration) per unit of carbon assimilated through photosynthesis], but even so, the net effect is a small decrease (~2%) in the available soil water in the Physiology scenario due to the simulated decrease in precipitation. This decrease is more pronounced in the Deforestation run (reaching a reduction of 30% at the peak of the dry season in September) because the GPP is considerably lower at this time of year in pasture grasslands, which, together with the lower evapotranspiration and the decreased input of rainwater, acts to decrease the soil water in the dry season in the Deforestation scenario (Fig. 6f).

## 4. Discussion

Our results show that the modelled responses to $eCO_2$ and large-scale deforestation are associated with equivalent reductions in the annual average precipitation and evapotranspiration in the Amazon region. The simulated decreases in precipitation (Physiology: 12%; Deforestation: 9%) are beyond the Amazon region rainfall interannual variability of 5% (Spracklen and Garcia-Carreras, 2015). Different climatological mechanisms drive such reductions in the two scenarios. Both scenarios have one mechanism behind the precipitation reduction in common: the reduced flux of moisture from surface vegetation to the atmosphere. The difference, however, is that in the Physiology scenario it is due to an $eCO_2$-driven reduction in the $g_s$ of forest vegetation, whereas in the Deforestation scenario it is due to a decrease in the leaf area index. Another similar mechanism of change in both scenarios is the alteration of the Walker cell over the Amazon: in the Physiology scenario, this occurs through a humidity-enriched free troposphere with decreased deep convection due to the heightening of a drier and warmer boundary layer, and in the Deforestation scenario, it occurs through a strengthened moisture convergence in the west/northwest Amazon and a subsidence branch over the east Amazon. On the other hand, different patterns of change in near-surface horizontal circulation imply substantial differences between the two scenarios with respect to the free-troposphere moisture content and 2-m temperature over the Amazon region.

In fact, the changes in the Walker circulation in the two scenarios take place for different reasons. In the deforestation scenario, the change is due to the strong intensification of the easterlies (Hadley Cell) across the Amazon and up to the Andes, driven specifically by the lower surface roughness length. In the Physiology scenario, two atmospheric circulation changes take place: on the one hand, the west winds from the Pacific are intensified, increasing precipitation over the Andes, especially in northern South America; on the other hand, the trade winds decrease (weakening of the Hadley Cell), which is apparently linked to a combination of a regional redistribution of convection and moisture convergence/divergence, changes

in the boundary layer depth and temperature, and, to a smaller extent, a teleconnection with $eCO_2$-driven climatic changes in tropical Africa, the latter of which was also shown by Kooperman et al. (2018). These results are corroborated by previous studies on the modelled effects of $eCO_2$ and deforestation on climate, though these previous studies used different models and model setups (i.e., they did not systematically compare the effects of both drivers using the same model(s) or followed a single modelling protocol). The combination of $eCO_2$ and deforestation (see Figs. S1 to S4 in the Supplement) results in

patterns for most of the variables that are similar to those obtained in the Deforestation scenario, except for the spatial pattern of rainfall change, which is less pronounced in the west Amazon, and for the circulation change pattern, in which the increase in easterlies in the west Amazon is not as strong as that in the Deforestation run, apparently due to the influence of β on atmospheric circulation over this region.

The Deforestation run conducted using CESM results in an equivalent precipitation reduction (-0.7 mm d$^{-1;}$ -12%) compared

to other studies that employed CESM/CLM to test the effects of $eCO_2$ on Amazon rainfall (Cao et al., 2010; Kooperman et al., 2018). However, the CESM simulation yields a different spatial pattern of rainfall change compared to the CPTEC-BAM run, with a stronger reduction/increase in precipitation in the east/west Amazon (Fig. S6), associated with a more pronounced strengthening of the Walker circulation and the cooling of the Amazon atmospheric column, as explained previously in the study by Badger & Dirmeyer (2016) using CESM. The rainfall change mechanisms are therefore similar

between the CPTEC-BAM and CESM runs.

### 4.1 Deforestation and rainfall in the Amazon

There is a long-known and overwhelming agreement among models that the whole-basin deforestation of the Amazon is associated with a warmer [average of 1.9°C [±1.8°C] vs. -0.2°C in 2-m air temperature (+0.4°C in surface temperature) in the current simulation with CPTEC-BAM] and drier (average -15% vs. -9% from CPTEC-BAM) climate over the region,

driven namely by an increase in trade winds due to the considerably smaller roughness length of pastures than that of forests (Lawrence and Vandecar, 2015; Sampaio et al., 2007; Spracklen and Garcia-Carreras, 2015; Sud et al., 1996). Fully interactive coupling between the atmosphere and oceans results in twice the rainfall reduction in comparison to that output by non-coupled simulation such as those conducted in the present study (Nobre et al., 2009). Although previous modelling and observational studies [e.g., (Saad et al., 2010; Silva Dias et al., 2002)] have shown that small-scale deforestation is

combined with a localized increase in rainfall, there is now modelling and observational evidence that widespread and large-scale deforestation in the Amazon drives rainfall reductions (Lawrence and Vandecar, 2015; Nobre et al., 2016; Sampaio et al., 2007) and/or the lengthening of the dry season (Dubreuil et al., 2012; Fu et al., 2013). This latter effect is also in line with our results (Fig. 6a).

While the conceptual model proposed/reviewed by Lawrence & Vandecar (2015) suggests that whole-basin deforestation

should lead to rainfall reductions of >30%, we argue that the longitudinal gradient in rainfall recycling should be considered in these estimates: the rainfall reductions observed with CPTEC-BAM in both the Deforestation and Physiology scenarios

are within the estimated range of precipitation recycling in the east Amazon [10% - 30% (Zemp et al., 2017)], which is the region where the subsidence branch of the Walker cell acts most strongly in these simulations.

## 4.2 $CO_2$ fertilization effect and moisture fluxes in the tropics

In contrast to the effect of deforestation on Amazon rainfall, observational or experimental evidence on the effects of $eCO_2$ on water fluxes in tropical forests is scarce. Most of the knowledge on the ecosystem-scale effects of $eCO_2$ comes from low-diversity temperate forests (Ainsworth and Long, 2005; Aisworth and Rogers, 2007; De Kauwe et al., 2013), laboratory studies with seedlings or saplings [e.g., (Aidar et al., 2002)], or growth rings obtained from trees at the fringes of tropical forests (van der Sleen et al., 2014). For example, the +150 ppm Oak Ridge free-air $CO_2$ enrichment (FACE) experiment

conducted in a broadleaf temperate forest resulted in an average reduction in transpiration of 17% (De Kauwe et al., 2013). A reduction of 20% in $g_s$ was found in the +150 ppm, single-species, eucalyptus FACE (EucFACE) experiment conducted in woodlands in New South Wales, Australia (Gimeno et al., 2016). Both results are comparable to the 18% reduction in $g_s$ and the 20% reduction in transpiration found in the Physiology scenario. However, water-use efficiency (calculated here as the ratio between GPP and transpiration) increased by 35% in the 11-year-long Oak Ridge FACE experiment and by 30-35% in

the 1850-2000 period, as assessed from growth rings from trees at the fringes of tropical forests (van der Sleen et al., 2014). Our simulation yielded a much higher value of 94% in the Physiology scenario, owing to a stronger increase in GPP in CPTEC-BAM (+13% in Oak Ridge FACE; +58% in CPTEC-BAM). Although the temperature dependence of Rubisco kinetics implies that the effects of $eCO_2$ on GPP and NPP in the tropics should, in principle, be stronger than those in temperate regions (Hickler et al., 2008), the GPP in CPTEC-BAM seems to be oversensitive to $eCO_2$, as is the case for other

vegetation models that do not consider nutrient cycling (De Kauwe et al., 2013). Phosphorus, for example, is a highly limiting nutrient in Amazon soils, and the consideration of such a limitation would decrease the expected $eCO_2$-induced gains in the GPP and NPP simulated by models without nutrient constraints by 42% and 50%, respectively, after 10 years (Fleischer et al. 2019). Observations from the strongly P-limited EucFACE site even showed a 12% increase in the GPP of mature *Eucalyptus tereticornis* stands after 4 years of $CO_2$ fertilization (Jiang et al., 2020). Should our simulations consider

the combined effect of P limitation, $g_s$ and therefore canopy transpiration would be even lower, and Amazon rainfall reduction could be even stronger in the Physiology scenario compared to that in the Deforestation scenario.

   One must also consider that in a hyper-diverse ecosystem such as the Amazon forest, the response to $eCO_2$ in terms of $g_s$ may vary considerably from one tree species to another or from one functional group/strategy of trees to another (Domingues et al., 2014). It is now known that different Amazon tree species can have rather different strategies regarding water usage

and saving (Bonal et al., 2000). Such a variety of responses and more subtle implications of $eCO_2$ on Amazon forest functioning have yet to be incorporated in vegetation models or surface schemes (Lapola, 2018).

   Therefore, even if our results for the Physiology scenario are aligned with observational results from non-tropical forest ecosystems and modelling results [namely, from the studies by Cao et al. (2010); Kooperman et al. (2018)], there is a considerable level of uncertainty in the Physiology scenario projection of CPTEC-BAM [and of CESM (Cao et al., 2010;

Kooperman et al., 2018)]. This level of uncertainty will stay as such until there are direct field-based data on the ecosystem-level effects of $eCO_2$ in the Amazon forest (Norby et al., 2016).

As such, we suggest that future research on this topic should focus on gathering such field-based experimental evidence on the ecosystem-level effects of $eCO_2$ in the Amazon forest and that the basin-wide effects of $eCO_2$ on Amazon rainfall should be projected with models that consider the potential limitations of soil phosphorus and interacting oceans. Additionally,
multi-factorial ensemble simulations with gradual increases in $CO_2$ concentrations and deforestation levels [*sensu* Sampaio et al., (2007)] could be valuable for understanding when and how the effects of increasing $CO_2$ and deforestation dominate the rainfall responses in the Amazon region. Last, the similarity of the results obtained for rainfall and evapotranspiration reduction with CPTEC-BAM under the $1.5xCO_2$ experiment and the results from CESM under the $2xCO_2$ (Cao et al., 2010) and $4xCO_2$ (Kooperman et al., 2018) scenarios might be a result, first, of the strong sensitivity of GPP and transpiration to
$eCO_2$ in CPTEC-BAM but could also be a consequence of the saturation of $eCO_2$ effects on $g_s$ that takes place between 600 and 1000 ppmv, as shown for a variety of plant species with instantaneous measurements [e.g., (Domingues et al., 2014; Zheng et al., 2019)], although the long-term (beyond the execution time of the FACE experiments) acclimation changes of $g_s$ to $eCO_2$ are still poorly known (Xu et al., 2016).

### 4.3 Mitigation perspectives

One should interpret the implications of the results presented here with care, keeping in mind the different responsibilities involved in the two anthropogenic disturbances considered in this modelling exercise: deforestation and elevated atmospheric $CO_2$ concentration. Avoiding the significant rainfall reductions projected here involves halting deforestation in the Amazon and reducing global $CO_2$ emissions or actively removing $CO_2$ from the atmosphere. On the one hand, the curbing of deforestation in the Amazon is something that invariably has to be carried out by different actors within the nine
Amazonian countries (France/French Guyana included), although international markets and institutions can play important roles as well (Nepstad et al., 2014; Rajão et al., 2020). On the other hand, the increase in atmospheric $CO_2$ concentration is a global process, the mitigation of which demands a concerted effort by all countries, especially the historical and current top emitters (Peters et al., 2015). In this sense, even if Amazon deforestation is stopped in the near future, forest functioning and structure can still be jeopardized by $eCO_2$ and consequent climatic changes. Therefore, while both anthropogenic
disturbances analysed in this study–deforestation and elevated atmospheric $CO_2$ concentrations–are associated with equivalent reductions in Amazon rainfall, this result should be interpreted as evidence that both regional and global responsibilities are at stake to mitigate potential future climatic change and its impacts in the region (Lapola et al., 2018).

### 5. Conclusions

In this study, we have, for the first time, applied a single coupled climate-vegetation model and standardized modelling
protocols to simulate the comparative impacts of the physiological (ß) effects of $eCO_2$ ($1.5xCO_2$) and large-scale (100%)

deforestation on precipitation in the Amazon region. Our results show equivalent decreases in the average annual precipitation for the two scenarios (Physiology or ß: 12%; Deforestation: 9%) that are well above the interannual variability in precipitation in the Amazon of 5%. The two scenarios also show reductions in the average annual evapotranspiration rates (Physiology or ß: -0.35 mm d$^{-1}$; Deforestation: -0.22 mm d$^{-1}$). Such a decreased input of moisture to the atmosphere is caused by an eCO$_2$-driven reduction in $g_s$ that is ultimately related to the 20% reduction in canopy transpiration in the Physiology scenario. In the Deforestation scenario, the reduction in the moisture flux from the vegetation to the atmosphere is related to the considerably lower leaf area index of pastures than that of forests. In both scenarios, changes are observed in the Walker circulation over tropical South America, with a convection zone concentrated over the Andes and weak subsidence over the east Amazon in the Deforestation scenario and a reduction in deep convection with high-troposphere subsidence anomalies in the Physiology scenario. However, the mechanisms driving such redistributions of convection within the Walker cell are different for each of the two scenarios. In the Physiology run, this effect is attributed to both the strengthening of west winds coming from the Pacific that increases rainfall in this region and is even associated with an increase in specific humidity over the free troposphere profile (this latter also related to a higher, warmer and drier boundary layer) and to the weakening of the Atlantic easterlies entering the Amazon basin due to the increased convection over Colombia and Venezuela and in tropical Africa. On the other hand, in the Deforestation scenario, this effect results from the considerable reduction in the surface roughness length that drives a strong increase in the easterlies flowing over the Amazon region, which is ultimately combined with the strengthening of Walker circulation. Our results for the Deforestation model run are in close agreement with those of previous observational and modelling studies. However, while our results for the Physiology scenario are at least partly aligned with observational studies conducted in non-tropical forests, data on growth rings from tropical trees, and other modelling studies, there is no direct, field-based experimental evidence on the ecosystem-level effects of eCO$_2$ on moisture fluxes (and other processes) in the Amazon forest, which confers a considerable level of uncertainty to these and other simulations on the ß effect of eCO$_2$ in the Amazon [e.g., (Kooperman et al., 2018)]. Overall, even if deforestation is completely stopped soon in the world's largest tropical forest, its climate system can still be jeopardized by eCO$_2$, ultimately depending on a process occurring in leaf stomata (Berry et al., 2010). Considering that the curbing of deforestation is a local/regional process (though it is tied to international markets and institutions) and that rising atmospheric CO$_2$ concentration is a global process, the reduction of which demands a concerted effort by all countries, it is clear that the responsibilities of Amazonian and non-Amazonian countries are at stake to mitigate the climatic changes projected here.

**Data availability**

The output data from CPTEC-BAM analysed here (last 10 years of each simulation) is deposited publicly at UNICAMP's Research Data Repository https://doi.org/10.25824/redu/OJMILK. Full-simulation data (100 years in the Physiology scenario and 30 years in the Deforestation scenario) and CPTEC-BAM source code is available upon reasonable request to the corresponding author.

## Author Contribution

GS, MC, CvR, LFR and DML designed the study; CAGJ, FA and MG carried out model runs and organized data curation; MS helped in the preparation of figures and analysis of data; MS, DML and GS prepared original manuscript draft; TD, AR, CvR, LFR and DML reviewed and edited earlier versions of the manuscript; GS and DML acquired funding; DML coordinated the project which this study is related to.

## Acknowledgements

This study is part of the AmazonFACE⅃ME project (https://labterra.cpa.unicamp.br/amazonface-me) and was funded through grants from Sao Paulo Research Foundation – FAPESP to DML (grant nº 2015/02537-7), CAGJ (grant nº 2017/07135-0), MC and GS (grant nº 2015/50122-0), LFCR (2017/03048-5) and CvR, GS and LFCR (2015/50687-8). GS and LFCR are also grateful to Brazil's National Council for Scientific and Technological Development – CNPq (grants nº 308158/2015-6 and 301084/2020-3) and TFD is thankful for USAID funding via the PEER program (grant nº AID-OAA-A-11-00012).

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

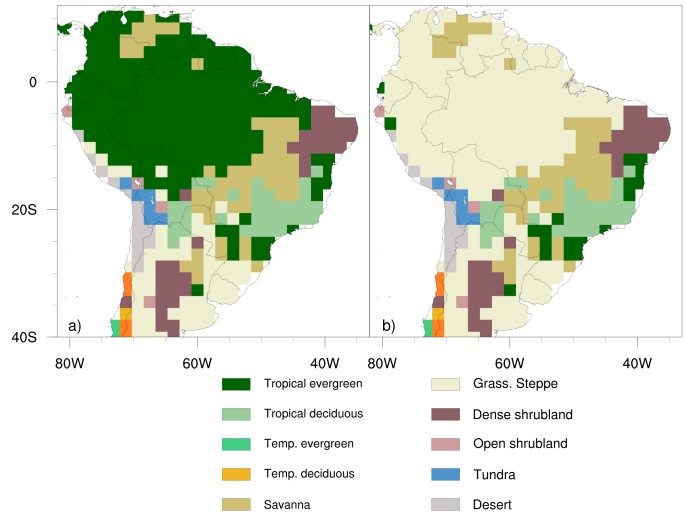

**Figure 1: Vegetation maps used in the (a) Physiology and (b) Deforestation modelling scenarios. The vegetation type of grass steppe in the Amazon region is composed of C$^4$ grass, representing tropical pasturelands.**

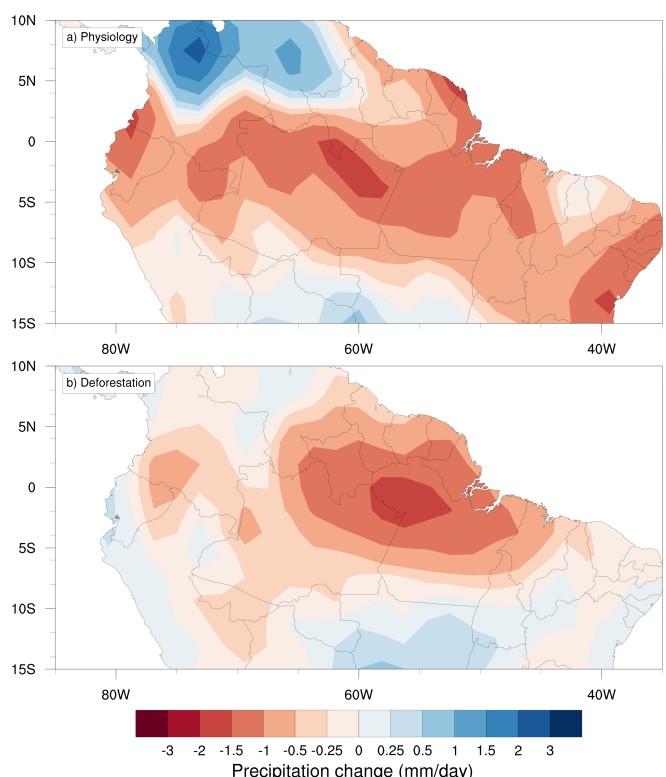

**Figure 2: Annual mean precipitation changes relative to control simulations obtained using CPTEC-BAM in tropical South America under (a) an atmospheric $CO_2$ concentration of +200 ppmv (1.5x$CO_2$) affecting solely surface vegetation physiology (Physiology) and (b) with the complete substitution of the Amazon forest by pasture grasslands and a control $CO_2$ concentration of 388 ppm (Deforestation).**


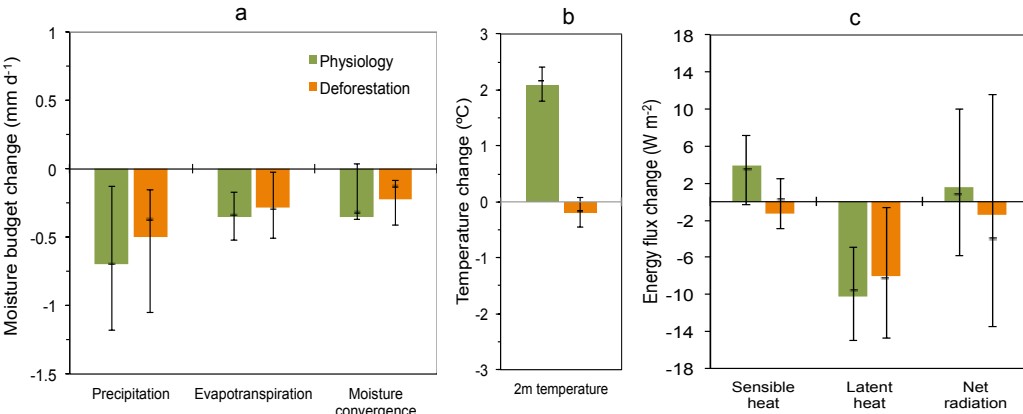


**Figure 3: Mean annual changes in (a) the moisture budget, (b) the 2-m air temperature and (c) the energy balance from the CPTEC-BAM over the Amazon region (black line square area in Fig. 5) under an atmospheric concentration of +200 ppmv (1.5xCO₂) affecting solely surface vegetation physiology (Physiology) and with the complete substitution of the Amazon forest by pasture grasslands (Deforestation). Solid lines indicate the interquartile range (25th, 50th and 75th percentile values) obtained**

**based on the spatial variability of the grid points used to determine the regional average.**

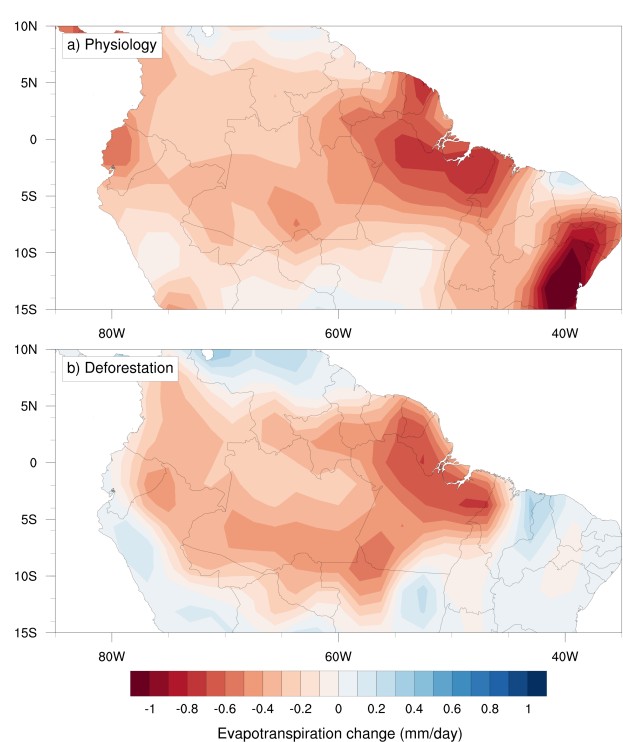

**Figure 4: Annual mean changes in evapotranspiration in tropical South America (a) under an atmospheric concentration of +200 ppmv (1.5xCO₂) affecting solely surface vegetation physiology and (b) with the complete substitution of the Amazon forest by**

**pasture grasslands.**

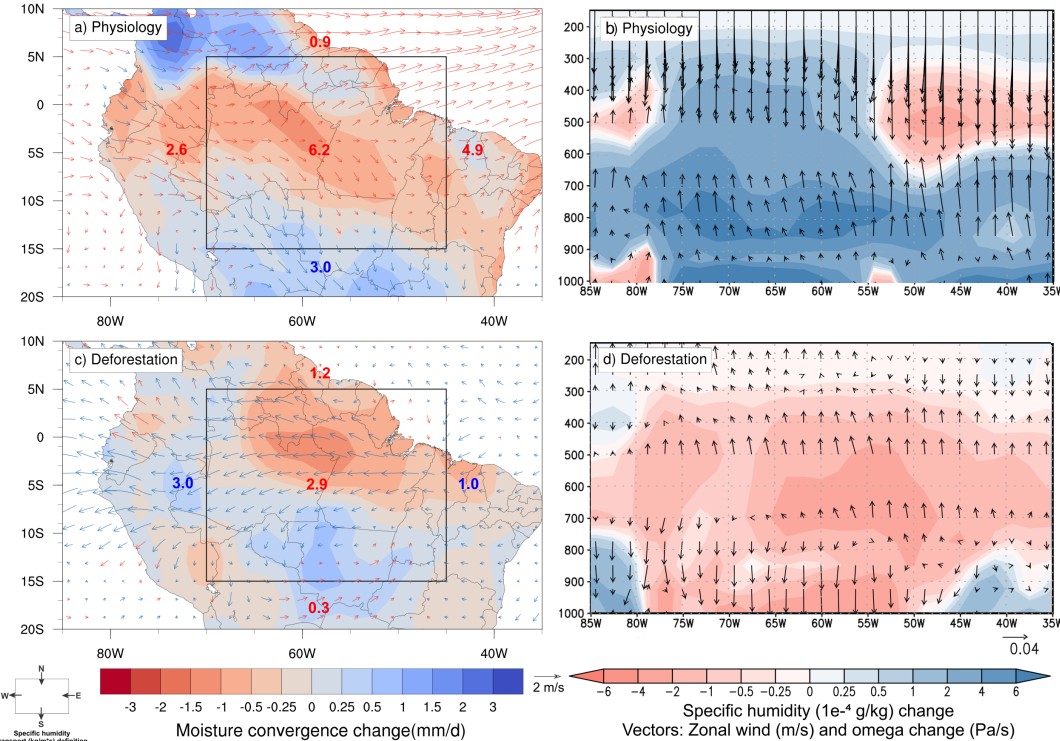

**Figure 5: Annual mean changes in the 850-hPa horizontal wind (a, c) and the vertical profile of zonal circulation over the equator superposed on meridional mean specific humidity vertical profile (with pressure in hPa as vertical coordinate) (b, d) in tropical South America under an atmospheric concentration of +200 ppmv (1.5xCO₂) (a, b) affecting solely surface vegetation physiology and (c, d) with the complete substitution of the Amazon forest by pasture grasslands. The black square depicts the region over the Amazon for which changes in the specific humidity flux balance (kg m⁻¹ s⁻¹, integrated up to 500 hPa) are calculated. The red and blue arrows/numbers represent decreases and increases, respectively, of the given variable.**

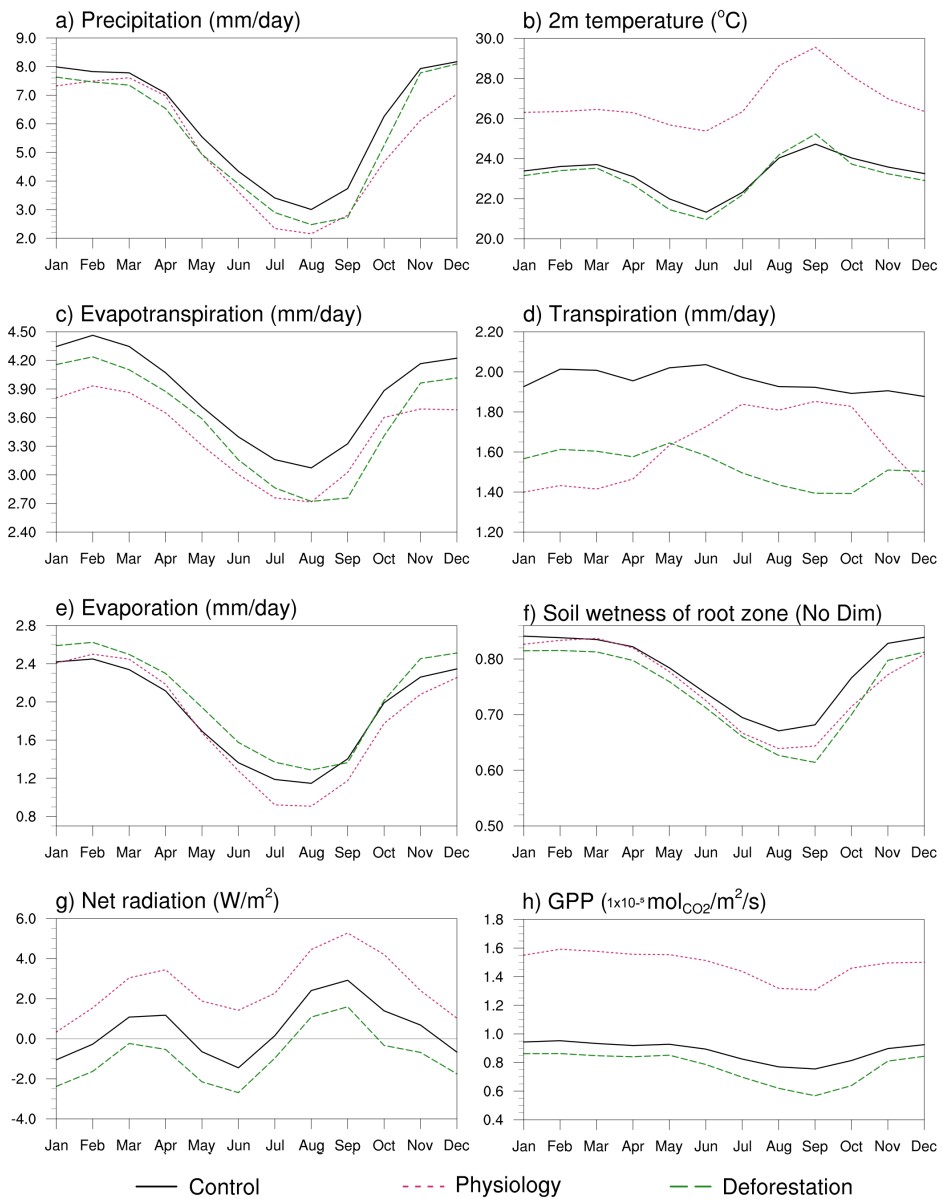


**Figure 6: Mean monthly precipitation (a), 2-m temperature (b), evapotranspiration (c), transpiration (d), evaporation (e), topsoil water content (f), net radiation (g) and gross primary productivity (h) in the Amazon region (black line square area in Fig. 5) in the Control, Physiology and Deforestation modelling scenarios.**


**Table 1: Numerical experiments performed with CPTEC-BAM.**

| Experiment | Vegetation | CO2 concentration (ppmv) | | Deforestation |
| --- | --- | --- | --- | --- |
| | | Atmosphere | Land Surface | |
| Control | Dynamic/Static* | 388 | 388 | No |
| Physiology | Dynamic | 388 | 588 | No |
| Deforestation | Static | 388 | 388 | Yes |
| RCP8.5+Def** | Dynamic | 588 | 588 | Yes |

*Control run with static vegetation was used for comparison with the Deforestation run.

** Results presented in Supplement.


**Table 2: Mean annual changes and interquartile ranges (25th, 50th and 75th percentile values in parentheses) of the moisture budget, 2-m air temperature, energy balance, GPP, $g_s$ and LAI from the CPTEC-BAM over the Amazon region (black line square area in Fig. 5) under an atmospheric concentration of +200 ppmv (1.5xCO$_2$) affecting solely surface vegetation physiology (Physiology) and with the complete substitution of the Amazon forest by pasture grasslands (Deforestation).**

| Variable \ Scenario | Physiology | Deforestation |
|---|---|---|
| Precipitation (mm d$^{-1}$) | -0.70 (-1.18; -0.70; -0.13) | -0.50 (-1.05; -0.37; -0.15) |
| Evapotranspiration (mm d$^{-1}$) | -0.35 (-0.52; -0.33; -0.17) | -0.28 (-0.51; -0.29; -0.02) |
| Transpiration (mm d$^{-1}$) | -0.35 (-0.53; -0.35; -0.19) | -0.42 (-0.66; -0.43; -0.19) |
| Moisture convergence (mm d$^{-1}$) | -0.35 (-0.37; -0.32; +0.04) | -0.22 (-0.41; -0.13; -0.08) |
| 2m temperature (ºC) | +2.07 (+1.80; +2.16; +2.40) | -0.20 (-0.45; -0.17; +0.08) |
| Sensible heat flux at surface (W m$^{-2}$) | +3.96 (-0.32; +3.46; +7.17) | -1.34 (-2.91; +0.23; +2.44) |
| Latent heat flux at surface (W m$^{-2}$) | -10.23 (-14.98; -9.60; -4.98) | -8.00 (-14.72; -8.27; -0.63) |
| Shortwave radiation at surface* (W m$^{-2}$) | +1.94 (+0.59; +2.23;+3.90) | +3.88 (+3.91; +5.08; +3.88) |
| Longwave radiation at surface* (W m$^{-2}$) | +2.75 (+2.24; +3.11; +3.85) | +6.9 (+4.84; +6.98; +9.26) |
| Net radiation (W m$^{-2}$) | -1.58 (-9.16; -0.79; +6.63) | +1.36 (-8.88; +4.02;+16.17) |
| Cloud cover (%) | -1.4 (-2.1; -1.4; -0.6) | -2.2 (-2.9; -2.3; -1.7) |
| Gross primary productivity ($\mu$molCO$_2$ m$^{-2}$ s$^{-1}$) | +7.0 (+5.0; +9.0; +9.0) | -1.0 (-2.0; -1.0; 0.0) |
| Stomatal conductance (molH$_2$O m$^{-2}$ s$^{-1}$) | -0.10 (-0.10; -0.07; -0.05) | -0.02 (-0.02; +0.001; +0.003) |
| Leaf area index | +10.0 (+7.0; +12.2; +13.2) | -4.1 (-5.5; -5.5; -2.7) |

*Balance between incoming/absorbed and reflected/emitted radiation

