# Peer review of "CO2 physiological effect can cause rainfall decrease as strong as large-scale deforestation in the Amazon"

_Biogeosciences, 2020_

## Referee Comment (RC1) · Anonymous Referee #1 · 7 Dec 2020

The authors used a coupled dynamic vegetation-atmosphere model to compare the impacts of the CO2 physiological effect and deforestation on Amazon precipitation. The results show that physiological forcing (x1.5CO2) and deforestation (forest->grassland) yield same amount of precipitation decreases in the Amazon, but the underlying mechanisms are different. This manuscript is well written and the topic is of interest to the biogeoscience community. To help further improve the manuscript, I have several suggestions below:

Major comments: This study compares the idealized physiological and deforestation simulations. In reality, both rising CO2 and deforestation are influencing precipita-

[Figure]

tion, so we are more interested in the compound effect of them. Although rising $CO_2$(x1.5$CO_2$) and deforestation reduce precipitation of a similar magnitude (12-13%), their mechanisms are different, and may amplify or attenuate each other. Here an interesting question arises: would the combination of rising $CO_2$(x1.5$CO_2$) and deforestation cause more or less than 25% of precipitation reductions? I am not sure how long it takes to run another scenario, but it is definitely worth a try.

Minor comments: Title: "$CO_2$ fertilization effect"->"$CO_2$ physiological effect"

Lines 138-140: temperature increases are due to reduced evaporative cooling effect in the physiological and deforestation scenarios, rather than precipitation decreases.

Lines 145-148: I think the logic here is that reductions in evapotranspiration and moisture convergence lead to precipitation decreases. More analyses of how land surface changes (physiology and deforestation) modify atmospheric processes and thereby impact moisture convergence and precipitation are needed here.

Section 3.1: add some statistical analyses of changes in stomatal conductance, leaf area index, transpiration, and atmospheric specific humidity in the physiological and deforestation scenarios.

Fig 5: To show how circulation changes impact moisture convergence, please also include moisture convergence changes in the physiological and deforestation scenarios in this figure.

Lines 196-198: as total evapotranspiration (transpiration+evaporation) is reduced, the decrease in soil water should not be due to increases in temperature and evaporation, but rather because of precipitation declines.

---

## Referee Comment (RC2) · Anonymous Referee #2 · 18 Dec 2020

In this study the authors assess the impacts of higher CO2 and deforestation on precipitation over the Amazon. They compare two simulations from the CPTEC-Brazilian Atmospheric Model, one with a 50% increase in the atmospheric CO2 concentration and another with all forest converted to grassland. The main results suggest that both scenarios lead to similar reductions in annual precipitation of 12% (higher CO2) and 13% (deforestation) due to equivalent reductions (20%) in transpiration. Though changes in precipitation and transpiration are similar, the causes (e.g., reduced stomatal conductance vs. reduced leaf area) and impacts on circulation differ.

Overall, this research addresses an important scientific question, with relevance to society. The focus on smaller increases in CO2, 1.5x vs 4x as in other work, is particularly relevant for the near future. The well referenced discussion section is also very useful for understanding how this work fits into previous literature. And the fact that precipitation changes are similar in the two simulations, but with very different circulation anomalies, is especially interesting. However, some claims are made without direct evidence/analysis to support them, and additional figures are need to clarify the mechanisms discussed. For example, a major conclusion is a strengthening of the Walker circulation in both scenarios, but only the 850mb wind field is shown. It would help to provide a much more robust analysis of circulation and moisture transport changes over the region. There are also some inconsistencies between the results in the figures and descriptions in the text. After addressing the comments below, this work would be a good fit with this journal.

Major Comments:

1. Is it a coincidence that 1.5xCO2 and 100% substitution to grassland give similar results? Why were these values chosen to compare? Why not 2xCO2 or 50% substitution to grassland? Some motivation for this specific comparison is needed. For example, when in the future would we expect to reach 1.5xCO2 based on the current trajectory of emissions, and how does that compare to the timescale of deforestation based on current deforestation rates?

2. The meridional mean changes in Figure 4c show increases in column specific humidity along the western side of the Amazon due to physiology, but the moisture transport in Figure 5a and discussion in the text indicate a reduction in the flux to the Andes (2.1 kg/m/s reduction). Likewise, in the deforestation case for the same region, there is a large reduction in low level humidity (Figure 4d), but Figure 5b shows an increase of 10.8 kg/m/s. In order to understand these results, it would be helpful to see what the horizontal wind anomalies look like at different levels? It might also help to decompose the moisture transport changes in order to understand the contribution from changes in humidity vs. changes in circulation.

3. Comparing Figures 6c and 6d, it seems that the evaporation from the soil and canopy are compensating the reduction from transpiration in the deforestation simulation during the wet season. It would be helpful to see what the seasonal cycle of these other evaporation terms look like? What would cause them to increase, despite a decrease in precipitation, in the deforestation simulation? It would also be helpful to see the seasonal cycle of LAI in this figure to better understand the mechanisms described. Likewise, adding annual mean net radiation to Figure 3c (and perhaps the season cycle of net radiation to Figure 6) would help clarify the mechanisms driving changes in the energy budget and surface temperature.

4. The fact that the two scenarios give similar results is really interesting. It is likely there would be some compensating effects on transpiration if they were simulated together (transpiration would decline by less than the two added together), but it is less clear what the impact on circulation would be. It would be very interesting to add a third scenario in which both elevated CO2 and deforestation occur together. There are several places in the manuscript that refer to a "Full" simulation which is not shown in the figures, is that referencing this combined scenario?

Minor Comments:

Line 26: What specifically is "its dry-season lower surface vegetation coverage" referring to aside from the "decreased leaf area index" that is already mentioned?

Line 80: Suggest changing "majorly in" to "mostly on".

Line 108: Does the vegetation component of CPTEC-BAM allow for increases in LAI, without a change vegetation type, due only to higher CO2? Or is it only the impact on stomatal conductance?

Line 110: Suggest changing to: "Control run with an atmospheric CO2 concentration of 388 ppmv."

Line 116: Should this be "climatological mean" or "climatological mean annual cycle"?

Often the seasonal cycle is retained in prescribed SST simulations, is that the case here?

Line 117: What does it mean that "vegetation distribution could vary"? Does this mean that a forest can become a grassland interactively? Does the model represent disturbances like drought and fires that are needed to drive this transition?

Line 135: What is the region averaged over in Figure 3? Is it the entire region shown in Figure 2 or just the Amazon? You could add a box to figure 1 or 2 showing the region. This should be stated in the figure caption as well.

Line 138: How do you know the "reduction in precipitation leads to an increase in regional temperature" and not the other way around? From the experiment, it seems more likely reduced ET leads to higher temperature and lower precipitation, and the temperature and precipitation changes likely feedback on each other. I suggest using language such as "is associated with" rather than "leads to". I suggest being careful about causal relationships throughout the manuscript.

Line 140: What is the 3rd scenario?

Line 143: It would be helpful to include the net surface radiation term as part of Figure 3c. Or even the up and down terms for short and longwave radiation. In the deforestation simulation, latent heat goes down, but sensible is mostly unchanged, which implies net radiation also goes down. Do you know why? Is it just due to the surface albedo change or are there changes that might impact other terms as well, such as cloud cover?

Line 147: Again, I don't think "yields" is a good word choice because it implies a causal relationship, which is not shown. I would think the "reduction of moisture convergence" yields "decreased precipitation" and not the other way around.

Line 153: What is the change in "gross primary productivity" in these simulations?

Line 157: What is the "Full" run mentioned here? "Full" is not a simulation mentioned

earlier or in Table 1.

Line 162: Figure 4 is not referred to in the text. It should be referenced somewhere before Figure 5.

Line 167: Could you show the impact of "decreased roughness length of surface" quantitatively? How does the boundary layer height change overall? What is the influence of roughness length vs. higher temperatures and heating on vertical mixing?

Line 171: Again, there is reference to a third scenario.

Line 173: What do the horizontal wind anomalies look like at different levels? It might help to decompose the moisture tranport changes in order to understand the contribution from changes in humidity vs. changes in wind.

Line 182: This section also references a third "Full" simulation not shown in the figures or described in the text/table.

Line 188: To understand the seasonality of the precipitation changes, and differences between wet and dry seasons, it would be useful to assess the circulation changes and moisture transport at a seasonal timescale as well. For instances, are the circulation changes due to eCO2 larger than deforestation during the rainy season, leading to a larger decrease in precipitation from Oct to Dec?

Line 189: I'm not sure the ET changes explain the moderate increase in temperature due to deforestation because it is more moderate throughout the year, including the dry season when ET decreases more than in the physiology simulation.

Line 198: Figure 6c shows a decrease in ET, but this sentence states that evaporation increases. I think the decrease in soil water in the physiology simulation is probably due to the decrease in precipitation, since ET decreases all year round.

Line 200: It would help to show the seasonal cycle of LAI or vegetation coverage in these simulations. That would help explain the seasonal pattern of ET and soil wetness

better, particularly for the deforestation run. This could be added as a panel to Figure 6.

Line 212: The "strengthening of the Walker cell over the Amazon" is not shown in the results. I suggest adding a figure that shows the full vertical-zonal wind anomalies.

---

## Author Comment (AC1) · 17 Feb 2021

We take this opportunity to genuinely thank the work done by the two anonymous reviewers, which has substantially improved this new version of our manuscript.

Obs: line numbers mentioned in the referee's comment refer to line numbers in the previous manuscript version, whereas the line numbers mentioned in the response to each comment refer to line numbers in the new manuscript PDF file.

IMPORTANT STATEMENT When attempting to respond to the referees' comments we realized that some of the variables requested to be shown in the article were not

saved or stored properly. As such we had to carry out new simulations such that we could properly save and show the requested variables. In that process we noticed that there were some differences in the Deforestation scenario results, especially regarding rainfall anomalies (which instead of -0.8 mm d-1 is in fact -0.5 mm d-1), and minor numerical updates in the other variables [although average precipitation reduction in the Physiology scenario is stronger than in the Deforestation scenario, the variability range of anomalies in both scenarios do not indicate a significant difference between the two mean values (as can be seen in Fig. 3a) and because of that we keep the article's title and conclusion]. We attribute this confusion regarding changing values in the Deforestation scenario to a recent substitution of processing blades at INPE's (Brazil's National Institute for Space Research) supercomputer, where these simulations were carried out. The new model runs do not change in any way the previous conclusions of the article and in fact are more trustworthy, for example in regard to the obtained changes in radiative balance and accompanying surface temperature changes which are now more consistent in the deforestation scenario. We sincerely apologize for the inconvenient.

**Anonymous Referee #1**

1. "This study compares the idealized physiological and deforestation simulations. In reality, both rising CO2 and deforestation are influencing precipitation, so we are more interested in the compound effect of them. Although rising CO2(x1.5CO2) and deforestation reduce precipitation of a similar magnitude (12-13%), their mechanisms are different, and may amplify or attenuate each other. Here an interesting question arises: would the combination of rising CO2(x1.5CO2) and deforestation cause more or less than 25% of precipitation reductions? I am not sure how long it takes to run another scenario, but it is definitely worth a try."

R: This point was also raised by Referee #2 and is indeed relevant for this article. One should notice however that if we have 100% deforestation and eCO2, the physiological effects of eCO2 would be acting upon grassland vegetation, and not on the forest any-
more, which was not the original aim of the article of assessing the comparative effects of the physiological effects of eCO2 on the forest and of an extreme deforestation scenario on rainfall in the Amazon region. In that sense, to not change the original concept of the article but attending the reviewers' suggestion we now present a scenario with eCO2 (in fact RCP8.5 which has a CO2 increase rate similar to what was employed in the Physiology and Deforestation scenarios) and 100% deforestation altogether as a supplement of this manuscript (Figs. S1-S4). In such a scenario (RCP8.5+Def) the changes in all variables are in between those obtained in the Deforestation and Physiology scenarios, except for the spatial pattern of rainfall change which is comparatively more pronounced in west Amazon; and also the circulation change pattern, in which the increase of Easterlies across the Amazon stronger than in Deforestation, apparently due to the combined effects of eCO2 on plant physiology and radiative balance of the atmosphere. Explicit mentions to this scenario and a brief discussion of its results in comparison to the Physiology and Deforestation scenarios are now made respectively in the main text lines 123 (Methods), and 307 (Discussion).

2. "Section 3.1: add some statistical analyses of changes in stomatal conductance, leaf area index, transpiration, and atmospheric specific humidity in the physiological and deforestation scenarios."

R: Statistics on stomatal conductance, leaf area index, transpiration and atmospheric specific humidity (this latter the atmospheric vertical profile over the Amazon) are now presented in section 3.1 (see lines 176-190), as well as in the newly included Table 2 (attached), that presents summarized statistics for all the variables analyzed in the article.

3. "Fig 5: To show how circulation changes impact moisture convergence, please also include moisture convergence changes in the physiological and deforestation scenarios in this figure."

R: We thank the reviewer for this very good suggestion. Fig. 5 now presents a spatially

**BGD**
explicit map of changes in moisture convergence overlaid by the anomalies in atmospheric circulation at 850 hPa (see Fig. 5 attached). We understand that the inclusion of such information in Fig. 5 makes it clearer the role of different circulation anomalies in driving the similar changes in the region's moisture budget. A mention in the sense is now made in line 206 (Results – 3.2 Atmospheric Circulation).

Other minor comments:

4. "Title: 'CO2 fertilization effect' -> 'CO2 physiological effect' "

R: Suggestion accepted.

5. "Lines 138-140: temperature increases are due to reduced evaporative cooling effect in the physiological and deforestation scenarios, rather than precipitation decreases."

R: Referee #1 is correct. It is the reduction in evapotranspiration that causes both the reduction in rainfall and increase in temperature. The mentioned sentence now reads (line 160):

"As expected for a tropical region where variations in precipitation and temperature are tightly coupled, the reduction in evaporative cooling leads to an increase in regional temperature (...)"

6. "Lines 145-148: I think the logic here is that reductions in evapotranspiration and moisture convergence lead to precipitation decreases. More analyses of how land surface changes (physiology and deforestation) modify atmospheric processes and thereby impact moisture convergence and precipitation are needed here."

R: We thank Referee #1 for indicating this point for improvement in the article. In fact the concept of moisture convergence employed here is a well-known simplification of the mass continuity equation applied to the specific humidity mass of an atmospheric volume:

S = P - E(1)

BGD
Where S is the storage of water vapour, P is precipitation and E is evapotranspiration (Banacos and Schultz 2005). We now make this information explicit in the article text when moisture convergence is first mentioned in the text and changed the phrasing to (line 171):

"The reduction of evapotranspiration (Physiology: -0.35 mm d-1; Deforestation: -0.22 mm d-1) is associated with a reduction of moisture convergence [precipitation minus evapotranspiration (Banacos and Schultz, 2005)] alongside with decreased precipitation in both Physiology and Deforestation model scenarios. Reduction in moisture convergence is 59% more pronounced in the Physiology scenario (Fig. 3a) owned namely to a stronger decrease in horizontal transport of humidity by east winds. The mechanisms associated with these changes are explained next."

While this specific section of text is just part of the opening of the Results section, the "analyses of how land surface changes (physiology and deforestation) modify atmospheric processes and thereby impact moisture convergence and precipitation" is provided in the subsequent subsections of the Results section, namely in subsections "3.1 Provision of humidity" and "3.2 Atmospheric circulation". In any case we make it explicit in the opening of the Results section that further explanation will be provided in the oncoming subsections (see line 175).

7. "Lines 196-198: as total evapotranspiration (transpiration+evaporation) is reduced, the decrease in soil water should not be due to increases in temperature and evaporation, but rather because of precipitation declines."

R: Referee #1 is correct and the mentioned sentence now reads (line 274):

"Stomatal closure driven by eCO2 is related to higher water use efficiency (the amount of water used [in transpiration] per unit of carbon assimilated through photosynthesis), but even so the net effect is a small decrease ( $\sim$ 2%) of available soil water in the Physiology scenario, due to the decrease in precipitation."

BGD
Please also note the supplement to this comment: https://bg.copernicus.org/preprints/bg-2020-386/bg-2020-386-AC1-supplement.pdf

**BGD**
**Fig. 1.** Vegetation maps used in (a) Physiology and (b) Deforestation modelling scenarios. Vegetation type grass. steppe in the Amazon region is composed of C4 grass, representing tropical pasturelands.

**BGD**

**Supplement:**

*Supplement of*

**CO₂ physiological effect can cause rainfall decrease as strong as large-scale deforestation in the Amazon**

**Gilvan Sampaio et al.**

*Correspondence to*: David M. Lapola (dmlapola@unicamp.br)

**S1. Supplementary figures**

[Figure]

**Fig. S1:** Annual mean precipitation change relative to control simulations using the CPTEC-BAM in tropical South America under an atmospheric $CO_2$ concentration of +150 ppmv (RCP8.5 in 2050) affecting both plant physiology and atmospheric radiative balance, and with concomitant complete substitution of the Amazon forest by pasture grasslands and a control $CO_2$ concentration of 388ppm (Deforestation).

[Figure]

**Fig. S2:** Annual mean changes in evapotranspiration (a) and meridional mean specific humidity vertical profile (with pressure in hPa as vertical coordinate) (b) in tropical South America under an atmospheric concentration of +150 ppmv (RCP8.5 in 2050) affecting both plant physiology and atmospheric radiative balance, and with concomitant complete substitution of the Amazon forest by pasture grasslands.

[Figure]

**Fig. S3:** Annual mean changes in 850 mb horizontal wind in tropical South America under an atmospheric concentration of +150 ppmv (RCP8.5 in 2050) affecting both plant physiology and atmospheric radiative balance, and with concomitant complete substitution of the Amazon forest by pasture grasslands. Black square depicts the region over the Amazon for which changes in the specific humidity flux balance (kg m-1 s-1, integrated up to 500hPa) is calculated. Red and blue arrows/numbers represent respectively decrease and increase of the given variable.

[Figure]

**Fig. S4:** Mean monthly precipitation, 2m-temperature, evapotranspiration, canopy transpiration and topsoil water content in the Amazon region (black line square in Fig. 5) in the control simulation and under an atmospheric concentration of +150 ppmv (RCP8.5 in 2050) affecting both plant physiology and atmospheric radiative balance, and with concomitant complete substitution of the Amazon forest by pasture grasslands.

[Figure]

**Fig. S5:** Meridional mean atmospheric boundary layer height over the Equator above on the Amazon region under different scenarios: green: Physiology; red: Deforestation; black: Physiology control; grey: Deforestation control.

[Figure]

**Figure S6:** Annual mean precipitation change relative to control simulation using CESM in tropical South America with complete substitution of the Amazon forest by pasture grasslands.

---

## Author Response (AR1)

**Response to 1st review round**

We take this opportunity to genuinely thank the work done by the two anonymous reviewers, which has substantially improved this new version of our manuscript.

*Obs: line numbers mentioned in the referee's comment refer to line numbers in the previous manuscript version, whereas the line numbers mentioned in the response to each comment refer to line numbers in the new manuscript PDF file (version without track changes).*

**IMPORTANT STATEMENT**
When attempting to respond to the referees' comments we realized that some of the variables requested to be shown in the article were not saved or stored properly. As such we had to carry out new simulations such that we could properly save and show the requested variables. In that process we noticed that there were some differences in the Deforestation scenario results, especially regarding rainfall anomalies (which instead of -0.8 mm d$^{-1}$ is in fact -0.5 mm d$^{-1}$), and minor numerical updates in the other variables [although average precipitation reduction in the Physiology scenario is stronger than in the Deforestation scenario, the variability range of anomalies in both scenarios do not indicate a significant difference between the two mean values (as can be seen in Fig. 3a) and because of that we keep the article's title and conclusion]. We attribute this confusion regarding changing values in the Deforestation scenario to a recent substitution of processing blades at INPE's (Brazil's National Institute for Space Research) supercomputer, where these simulations were carried out. The new model runs do not change in any way the previous conclusions of the article and in fact are more trustworthy, for example in regard to the obtained changes in radiative balance and accompanying surface temperature changes which are now more consistent in the deforestation scenario. We sincerely apologize for the inconvenient.

**Anonymous Referee #1**

1. "*This study compares the idealized physiological and deforestation simulations. In reality, both rising $CO_2$ and deforestation are influencing precipitation, so we are more interested in the compound effect of them. Although rising $CO_2$(x1.5$CO_2$) and deforestation reduce precipitation of a similar magnitude (12-13%), their mechanisms are different, and may amplify or attenuate each other. Here an interesting question arises: would the combination of rising $CO_2$(x1.5$CO_2$) and deforestation cause more or less than 25% of precipitation reductions? I am not sure how long it takes to run another scenario, but it is definitely worth a try.*"
R: This point was also raised by Referee #2 and is indeed relevant for this article. One should notice however that if we have 100% deforestation and e$CO_2$, the physiological effects of e$CO_2$ would be acting upon grassland vegetation, and not on the forest anymore, which was not the original aim of the article of assessing the comparative effects of the physiological effects of e$CO_2$ on the forest and of an extreme deforestation scenario on rainfall in the Amazon region. In that sense, to not change the original concept of the article but attending the reviewers' suggestion we now present a scenario with e$CO_2$ (in fact RCP8.5 which has a $CO_2$ increase rate similar to what was employed in the Physiology and Deforestation scenarios) and 100% deforestation altogether as a supplement of this manuscript (Figs. S1-S4). In such a scenario (RCP8.5+Def) the changes in all variables are in between those obtained in the Deforestation and Physiology scenarios, except for the spatial pattern of rainfall change which is comparatively more pronounced in west Amazon; and also the circulation change pattern, in which the increase of Easterlies across the Amazon stronger than in Deforestation, apparently due to the combined effects of eCO2 on plant physiology and radiative balance of the atmosphere. Explicit mentions to this scenario and a brief discussion of its results in comparison to the Physiology and Deforestation scenarios are now made respectively in the main text lines 123 (Methods), and 291 (Discussion).

2. "*Section 3.1: add some statistical analyses of changes in stomatal conductance, leaf area index, transpiration, and atmospheric specific humidity in the physiological and deforestation scenarios.*"
R: Statistics on stomatal conductance, leaf area index, transpiration and atmospheric specific humidity (this latter the atmospheric vertical profile over the Amazon) are now presented in section 3.1 (see lines 173-185), as well as in the newly included Table 2, that presents summarized statistics for all the variables analyzed in the article.

3. "*Fig 5: To show how circulation changes impact moisture convergence, please also include moisture convergence changes in the physiological and deforestation scenarios in this figure.*"
R: We thank the reviewer for this very good suggestion. Fig. 5 now presents a spatially explicit map of changes in moisture convergence overlaid by the anomalies in atmospheric circulation at 850 hPa. We understand that the inclusion of such information in Fig. 5 makes it clearer the role of different circulation anomalies in driving the similar changes in the region's moisture budget. A mention in this sense is now made in line 204 (Results – 3.2 Atmospheric Circulation).

Other minor comments:

4. "*Title: 'CO$_2$ fertilization effect' -> 'CO$_2$ physiological effect'* "
R: Suggestion accepted.

5. "*Lines 138-140: temperature increases are due to reduced evaporative cooling effect in the physiological and deforestation scenarios, rather than precipitation decreases.*"
R: Referee #1 is correct. It is the reduction in evapotranspiration that causes both the reduction in rainfall and increase in temperature. The mentioned sentence now reads (line 159):

> "As expected for a tropical region where variations in precipitation and temperature are tightly coupled, reductions in evaporative cooling and changes in atmospheric circulation are combined with changes in the regional near-surface air temperature (...)"

6. "*Lines 145-148: I think the logic here is that reductions in evapotranspiration and moisture convergence lead to precipitation decreases. More analyses of how land surface changes (physiology and deforestation) modify atmospheric processes and thereby impact moisture convergence and precipitation are needed here.*"
R: We thank Referee #1 for indicating this point for improvement in the article. In fact the concept of moisture convergence employed here is a well-known simplification of the mass continuity equation applied to the specific humidity mass of an atmospheric volume:

$$S = P - E \qquad (1)$$

Where S is the storage of water vapour, P is precipitation and E is evapotranspiration (Banacos and Schultz 2005). We now make this information explicit in the article text when moisture convergence is first mentioned in the text and changed the phrasing to (line 167):

> "The reductions in evapotranspiration (Physiology: -0.35 mm d$^{-1}$; Deforestation: -0.28 mm d$^{-1}$) are associated with reductions in moisture convergence [precipitation minus evapotranspiration (Banacos and Schultz, 2005)] alongside decreased precipitation in both the Physiology and Deforestation model scenarios. The reduction in moisture convergence is 59% more pronounced in the Physiology scenario (Fig. 3a) than in the Deforestation scenario due to the strong reduction in the horizontal transport

> of humidity by easterly winds. The mechanisms associated with these changes are explained in the next sections."

While this specific section of text is just part of the opening of the Results section, the "analyses of how land surface changes (physiology and deforestation) modify atmospheric processes and thereby impact moisture convergence and precipitation" is provided in the subsequent subsections of the Results section, namely in subsections "3.1 Provision of humidity" and "3.2 Atmospheric circulation". In any case we make it explicit in the opening of the Results section that further explanation will be provided in the oncoming subsections (see line 172).

7. "*Lines 196-198: as total evapotranspiration (transpiration+evaporation) is reduced, the decrease in soil water should not be due to increases in temperature and evaporation, but rather because of precipitation declines.*"
R: Referee #1 is correct and the mentioned sentence now reads (line 261):

> "Stomatal closure driven by $eCO_2$ is related to higher water use efficiency (the amount of water used [in transpiration] per unit of carbon assimilated through photosynthesis), but even so, the net effect is a small decrease (~2%) in the available soil water in the Physiology scenario due to the simulated decrease in precipitation."

**Reviewer #2**

Major Comments:

1. "*Is it a coincidence that 1.5xCO2 and 100% substitution to grassland give similar results? Why were these values chosen to compare? Why not 2xCO2 or 50% substitution to grassland? Some motivation for this specific comparison is needed. For example, when in the future would we expect to reach 1.5xCO2 based on the current trajectory of emissions, and how does that compare to the timescale of deforestation based on current deforestation rates?*"
R: The thank Refereee #2 for point out this caveat in our manuscript and providing us the opportunity to make it clearer in that regard. The following text has been added to the Methods' subsection 2.2 Modeling protocol (line 126):

> "The selection of such scenarios starts with the intention of understanding the impacts on moisture fluxes and rainfall in the Amazon that are driven by the target concentration to be used in the AmazonFACE experiment in the central Amazon (Norby et al., 2016). Second, we also wanted to know how the results obtained in the Physiology scenario compared to the changes expected due to extreme deforestation in the region. Rather than representing realistic projections of the future of the Amazon, this systematic separation of climatic forcing types allows us to better understand how each forcing contributes to future changes in the region. Notwithstanding, an atmospheric $CO_2$ concentration of +200 ppm (i.e., 588 ppm) is projected to be reached shortly after 2050 under the IPCC RCP8.5 scenario and in 2080 under the RCP6.0 scenario (Vuuren et al., 2011). Complete deforestation of the Amazon basin, following a business-as-usual deforestation-rate scenario–with deforestation rates typical of the late 1990s–could possibly be reached in approximately 2100 (Soares-Filho et al., 2006)."

Moreover, we have added a sentence at the end of the manuscript that it would be probably valuable to perform ensemble simulations with gradual increase of $CO_2$ and deforestation levels, to understand when and how the effects of increasing $CO_2$ and deforestation dominate the rainfall responses in the Amazon region (see line 356).

2. "*The meridional mean changes in Figure 4c show increases in column specific humidity along the western side of the Amazon due to physiology, but the moisture transport in Figure 5a and discussion in the text indicate a reduction in the flux to the Andes (2.1 kg/m/s reduction). Likewise, in the deforestation case for the same region, there is a large reduction in low level humidity (Figure 4d), but Figure 5b shows an increase of 10.8 kg/m/s. In order to understand these results, it would be helpful to see what the horizontal wind anomalies look like at different levels? It might also help to decompose the moisture transport changes in order to understand the contribution from changes in humidity vs. changes in circulation.*"
R: This is a well-noticed point and we thank the reviewer for grating us the opportunity to explain this better as follow (line 209, section 3.2 Atmospheric circulation):

> "These changes in horizontal circulation imply, in the Physiology scenario, that less moisture enters the Amazon region from the Atlantic (-4.9 kg m$^{-1}$ s$^{-1}$) and less moisture leaves the regions towards the Andes (-2.6 kg m$^{-1}$ s$^{-1}$) (this latter is somewhat compensated by a stronger moisture convergence from the Pacific to the Andes, as shown in Fig. 5b). In the Deforestation scenario, there is an increase in the input of humidity to the Andes at the surface level (on the order of 3.0 kg m$^{-1}$ s$^{-1}$), which is also perceptible in the western part of the vertical humidity profile near the surface levels (Fig. 5d). The lower evapotranspiration capacity aligned with the lower vertical mixing due to

pasture's lower roughness length (than that of forests) results in an atmospheric volume that is depleted of moisture and shows a decreased uplifting of air masses. In the Physiology scenario, despite the decreased evapotranspiration capacity, the increased surface heating increases vertical mixing at low levels (up to 700 hPa), associated with a deeper boundary layer and higher mixing layer, which is, in turn, connected to the increase in humidity throughout the free tropospheric volume (above the boundary layer) over the region. However, after such atmospheric heights, there are strong subsidence anomalies seen in the Physiology run (Fig. 5b), which decrease deep convection that is ultimately associated with lower rainfall rates. The same vertical circulation patterns have been demonstrated well in previous (separate) studies that modelled the large-scale deforestation of the Amazon and, more recently, the isolated physiological effects of $eCO_2$ on the region's climate (*c.f.* Langenbrunner et al. 2019)."

3. "*Comparing Figures 6c and 6d, it seems that the evaporation from the soil and canopy are compensating the reduction from transpiration in the deforestation simulation during the wet season. It would be helpful to see what the seasonal cycle of these other evaporation terms look like? What would cause them to increase, despite a decrease in precipitation, in the deforestation simulation? It would also be helpful to see the seasonal cycle of LAI in this figure to better understand the mechanisms described. Likewise, adding annual mean net radiation to Figure 3c (and perhaps the season cycle of net radiation to Figure 6) would help clarify the mechanisms driving changes in the energy budget and surface temperature.*"
R: The reviewer is correct. Indeed evaporation is compensating the lower transpiration during the wet season in the Deforestation run, as is now clearly evidenced in Figure 6 and also mentioned in the text of section 3.4 Seasonality (line 258). Notice however that evapotranspiration, evaporation and transpiration are all lower than the control run. While including the seasonal cycle of LAI in Figure 6 is a good suggestion, we opted to not include there in this figure because we found out that there is no seasonal variation of LAI in the vegetation scheme of CPTEC-BAM. The corresponding average LAI changes for each run are now provided in the main text (line 181) and in the newly included Table 2. Annual mean net radiation has been included in Fig. 3c, as well as a panel showing net radiation seasonal cycle in Fig. 6. The results related to the radiation balance are now discussed in an own section (3.3 Radiation balance) on lines 225-239. See also the response to Referee #2's comment 14.

4. "*The fact that the two scenarios give similar results is really interesting. It is likely there would be some compensating effects on transpiration if they were simulated together (transpiration would decline by less than the two added together), but it is less clear what the impact on circulation would be. It would be very interesting to add a third scenario in which both elevated CO2 and deforestation occur together. There are several places in the manuscript that refer to a "Full" simulation which is not shown in the figures, is that referencing this combined scenario?*"
R: Please see the response to Referee #1's comment 1.

Minor comments:

5. "*Line 26: What specifically is "its dry-season lower surface vegetation coverage" referring to aside from the "decreased leaf area index" that is already mentioned?*"
R: Indeed, the phrasing was redundant and now is restricted to "(…) smaller leaf area index in Deforestation" (see line 27).

6. "*Line 80: Suggest changing "majorly in" to "mostly on".*"
R: Suggestion accepted (see line 85).

7. "*Line 108: Does the vegetation component of CPTEC-BAM allow for increases in LAI,*"

*without a change vegetation type, due only to higher CO2? Or is it only the impact on stomatal conductance?*"
R: Yes, the vegetation component of CPTEC-BAM allows for increases in LAI, without a full change in the vegetation type. For sake of clarity we have rephrased the mentioned sentences to (line 115):

> "The numerical experiments employed here include simulations that consider the increase in the concentration of atmospheric $CO_2$ affecting plant physiology as well as experiments that consider deforestation in the Amazon, as follows (…)"

8. "*Line 110: Suggest changing to: "Control run with an atmospheric $CO_2$ concentration of 388 ppmv."*"
R: Suggestion accepted. The sentence now also includes complementary information about the way vegetation is modeled in the control runs to be compared with Physiology and Deforestation scenarios (line 117):

> "Control runs with an atmospheric $CO_2$ concentration of 388 ppmv, one with a dynamic and another with a static geographical distribution of vegetation types (for comparison with the Physiology and Deforestation scenarios, respectively)."

9. "*Line 116: Should this be "climatological mean" or "climatological mean annual cycle"? Often the seasonal cycle is retained in prescribed SST simulations, is that the case here?*"
R: Sea surface temperature (SST) was represented as the fixed climatological mean annual cycle from the period of 1981-2010 and, as such, it considered SST seasonal cycle. The information was included in the text (line 135).

10. "*Line 117: What does it mean that "vegetation distribution could vary"? Does this mean that a forest can become a grassland interactively? Does the model represent disturbances like drought and fires that are needed to drive this transition?*"
R: We meant that the geographical distribution of vegetation types can change according to variations of climate. The model simulates both grass and trees PFTs coexisting in a grid cell and disturbances such as fire are represented by a fixed percentage of the biomass of all PFTs that is reduced every year (this information is now included in line 91). So, depending on the environmental conditions (e.g. climate), in extreme cases, a forest could indeed become grassland. Nevertheless, as mentioned in the same sentence, there are no significant changes of vegetation in the Physiology scenario that are worth analysis in the article. Anyhow, the information is kept for sake of reproducibility [not without specifying that it is the "geographical distribution of vegetation types could vary throughout the model run (…)" (see line 136)] .

11. "*Line 135: What is the region averaged over in Figure 3? Is it the entire region shown in Figure 2 or just the Amazon? You could add a box to figure 1 or 2 showing the region. This should be stated in the figure caption as well.*"
R: In fact there is a mention in that sense Fig. 3 caption (line 606):

> "(…) from the CPTEC-BAM over the Amazon region (black line square area in Fig. 5)…"

12. "*Line 138: How do you know the "reduction in precipitation leads to an increase in regional temperature" and not the other way around? From the experiment, it seems more likely reduced ET leads to higher temperature and lower precipitation, and the temperature and precipitation changes likely feedback on each other. I suggest using language such as "is associated with" rather than "leads to". I suggest being careful about causal relationships throughout the manuscript.*"

R: The suggestion is pertinent and we have accepted it throughout the manuscript, giving preference to terms like "is associated with" or "is combined with" instead of "leads to" or "causes". Regarding the relation between ET, temperature and precipitation, please see the response to Referee #1's comment 5.

13. "*Line 140: What is the 3rd scenario?*"
R: Please see the response to Referee #1's comment 1. We try to restrict mentions only to Physiology and Deforestation scenarios, as these compose the main focus of the article.

14. "*Line 143: It would be helpful to include the net surface radiation term as part of Figure 3c. Or even the up and down terms for short and longwave radiation. In the deforestation simulation, latent heat goes down, but sensible is mostly unchanged, which implies net radiation also goes down. Do you know why? Is it just due to the surface albedo change or are there changes that might impact other terms as well, such as cloud cover?*"
R: Net surface radiation is now included in Figure 3c. The updated model runs now show that in fact there is a decrease also of sensible heat (-1.34 W m$^{-2}$) in the deforestation run, resulting in negative net surface radiation balance in the deforestation run, that is associated with a small decrease in average 2m-air temperature. The following text now composes a new article's subsection (3.3 Radiation balance) starting in line 225:

> "**3.3 Radiative balance**
> A decrease in the surface sensible heat (-1.34 W m$^{-2}$) in the Deforestation run (Fig. 3c), alongside a decrease in the latent heat, results in a negative net surface radiation balance in the Deforestation run, associated with a small decrease in the average 2-m air temperature (-0.2°C) (Table 2) (but also with an increase of +0.4°C in surface temperature). On the other hand, in the Physiology scenario, an increase in sensible heat (+3.96 W m$^{-2}$) is observed, associated with an average increase in the 2-m air temperature of +2.1°C. While the decrease in latent heat is also directly connected to a lower evapotranspiration capacity, the opposite results shown in each scenario regarding sensible heat are also associated with opposite changes in near-surface atmospheric circulation patterns: in the Deforestation run, there is an increase in near-surface atmospheric advection, whereas in the Physiology scenario, this advection is considerably decreased (as explained in section 3.2 Atmospheric circulation). Shortwave radiation is increased due to decreased nebulosity in both model scenarios (Physiology: -1.4%; Deforestation: -2.2%), but such an increase in the shortwave radiation balance is stronger in the Deforestation scenario due to the albedo change. The same pattern is also obtained for the surface balance of longwave radiation, which increases in both scenarios but increases more strongly in the Deforestation run (Physiology: 2.7 W m$^{-2}$; Deforestation: 6.9 W m$^{-2}$), which is probably a combination of the lower evapotranspiration capacity and increased horizontal advection in the latter scenario."

15. "*Line 147: Again, I don't think "yields" is a good word choice because it implies a causal relationship, which is not shown. I would think the "reduction of moisture convergence" yields "decreased precipitation" and not the other way around.*"
R: We thank Referee #2's patience in also noting this point. We have substitute the term "yields" by "is associated with". The phrasing of the mentioned sentence is now changed as shown in the response to Referee #1's comment 6.

16. "*Line 153: What is the change in "gross primary productivity" in these simulations?*"
R: The changes in GPP are now shown in Fig. 6, Table 2 and in the presentation of results in the main text (line 176):

> "The effect that a higher $CO_2$ concentration has on reducing $g_s$ (Eq. 1)

overcomes the positive effect of increased gross primary productivity (GPP) (Physiology: +7.0 $\mu molCO_2$ $m^{-2}$ $s^{-1}$ (+58%); Deforestation: -1.0 $\mu molCO_2$ $m^{-2}$ $s^{-1}$ (-16%) on $g_s$ (…)"

17. "*Line 157: What is the "Full" run mentioned here? "Full" is not a simulation mentioned earlier or in Table 1.*"
R: Please see the response to Referee #1's comment 1. Table 1 now explicitly includes the RCP8.5+Def scenario.

18. "*Line 162: Figure 4 is not referred to in the text. It should be referenced somewhere before Figure 5.*"
R: Two mentions to Figure 4 are now made in section "3.1 Provision of humidity" (lines 176 and 180).

19. "*Line 167: Could you show the impact of "decreased roughness length of surface" quantitatively? How does the boundary layer height change overall? What is the influence of roughness length vs. higher temperatures and heating on vertical mixing?*"
R: We now include a figure in the article's supplement (Fig. S5) showing the meridional mean planetary boundary layer height at the equator over the Amazon. We see that in the Deforestation scenario there is an average decrease of 10% in the boundary layer height, attributable to the considerably lower surface roughness length of pasture compared to a tropical forest. On the other hand, there is an average increase of 21% in boundary layer height in the Physiology run, associated with the increased heating of surface. This explanation is now given on main text line 196.

20. "*Line 171: Again, there is reference to a third scenario.*"
R: Please see Referee #1's comment 1. We have excluded the mention to the third scenario given that Physiology and Deforestation scenarios are the main focus of this article. Additionally, we make now an explicit reference to Fig. 3c and Table 2 in this specific part of the text (line 207) to help the reader find the results linked to the information provided in the mentioned sentence.

21. "*Line 173: What do the horizontal wind anomalies look like at different levels? It might help to decompose the moisture transport changes in order to understand the contribution from changes in humidity vs. changes in wind.*"
R: Please see the response to Referee #2's comment 2 above.

22. "*Line 182: This section also references a third "Full" simulation not shown in the figures or described in the text/table.*"
R: Please see Referee #1's comment 1. Additionally, we make now an explicit reference to Fig. S4a to help the reader find the results linked to this information (line 242).

23. "*Line 188: To understand the seasonality of the precipitation changes, and differences between wet and dry seasons, it would be useful to assess the circulation changes and moisture transport at a seasonal timescale as well. For instances, are the circulation changes due to eCO_2 larger than deforestation during the rainy season, leading to a larger decrease in precipitation from Oct to Dec?*"
R: While we agree with the Referee that the seasonality of changes is important, we do not include such an analysis of the seasonal changes of atmospheric circulation and moisture transport because it has been already shown, for example by Kooperman et al. (2018) that the Physiological effects of $eCO_2$ on the region's climate take place namely in the wet season, when GPP and gs/transpiration are higher (see Fig. 6d and h), even though our results also show considerable rainfall reduction during the dry season. Conversely, it has been demonstrated (e.g. Lawrence & Vandecar 2015) that large-scale deforestation causes climatic changes namely during the dry season, when transpiration is particularly reduced, as also shown in our results (Fig. 6a and d). This explanation is

now provided in the main text's section 3.4 Seasonality (lines 255-266).

24. "*Line 189: I'm not sure the ET changes explain the moderate increase in temperature due to deforestation because it is more moderate throughout the year, including the dry season when ET decreases more than in the physiology simulation.*"
R: The reviewer is correct. The sentence has been eliminated. A new sentence has been included in the previous section (line 202) that we believe explains better the mechanism behind the moderate increase in temperature in the Deforestation scenario:

> "On the other hand, the strong increase in westward moisture advection, aligned with the increased albedo and decreased vertical mixing (Fig. S5) seems to best explain the nearly unchanged surface temperature seen in the Deforestation scenario."

25. "*Line 198: Figure 6c shows a decrease in ET, but this sentence states that evaporation increases. I think the decrease in soil water in the physiology simulation is probably due to the decrease in precipitation, since ET decreases all year round.*"
R: Please see the response to Referee #1's point 7.

26. "*Line 200: It would help to show the seasonal cycle of LAI or vegetation coverage in these simulations. That would help explain the seasonal pattern of ET and soil wetness better, particularly for the deforestation run. This could be added as a panel to Figure 6.*"
R: Please see the response to Referee #2's comment 3 above.

27. "*Line 212: The "strengthening of the Walker cell over the Amazon" is not shown in the results. I suggest adding a figure that shows the full vertical-zonal wind anomalies.*"
`R: Please see the response to Referee #2's comment 2 above

**References**

Banacos, P. C. and Schultz, D. M.: The Use of Moisture Flux Convergence in Forecasting Convective Initiation: Historical and Operational Perspectives, Weather Forecast., 20(3), 351–366, doi:10.1175/WAF858.1, 2005.
Kooperman, G. J., Chen, Y., Hoffman, F. M., Koven, C. D., Lindsay, K., Pritchard, M. S., Swann, A. L. S. and Randerson, J. T.: Forest response to rising CO2 drives zonally asymmetric rainfall change over tropical land, Nat. Clim. Chang., 8(5), 434–440, doi:10.1038/s41558-018-0144-7, 2018.
Langenbrunner, B., Pritchard, M. S., Kooperman, G. J. and Randerson, J. T.: Why Does Amazon Precipitation Decrease When Tropical Forests Respond to Increasing CO2?, Earth's Futur., 7(4), 450–468, doi:10.1029/2018EF001026, 2019.
Lawrence, D. and Vandecar, K.: Effects of tropical deforestation on climate and agriculture, Nat. Clim. Chang., 5(1), 27–36, doi:10.1038/nclimate2430, 2015.
Norby, R. J., De Kauwe, M. G., Domingues, T. F., Duursma, R. A., Ellsworth, D. S., Goll, D. S., Lapola, D. M., Luus, K. A., MacKenzie, A. R., Medlyn, B. E., Pavlick, R., Rammig, A., Smith, B., Thomas, R., Thonicke, K., Walker, A. P., Yang, X. and Zaehle, S.: Model-data synthesis for the next generation of forest free-air CO 2 enrichment (FACE) experiments, New Phytol., 209(1), 17–28, doi:10.1111/nph.13593, 2016.
Soares-Filho, B. S., Nepstad, D. C., Curran, L. M., Cerqueira, G. C., Garcia, R. A., Ramos, C. A., Voll, E., McDonald, A., Lefebvre, P. and Schlesinger, P.: Modelling conservation in the Amazon basin, Nature, 440(7083), 520–523, doi:10.1038/nature04389, 2006.
Vuuren, D. P. Van, Edmonds, J., Kainuma, M., Riahi, K., Nakicenovic, N., Smith, S. J. and Rose, S. K.: The representative concentration pathways : an overview, Climati, 5–31, doi:10.1007/s10584-011-0148-z, 2011.